# Investigations of a new EPS-insulated mortar-free reusable interlocking block for sustainable buildings

Hui Gao, Moses Mwenda Karani*, Li Zuo, Zhongwei Zhao

School of Civil Engineering, Liaoning Technical University, Fuxin, China

* moseskarani124@gmail.com

## Abstract

Sustainable, energy-efficient, and reusable construction systems have increasingly been demanded due to the increasing amount of plastics, waste, demolition debris, and the carbon footprints associated with existing traditional materials used in the construction industry. In contribution to this, this study innovates a new enveloped insulated interlocking concrete block (IICB) made of recycled aggregate concrete (RAC) and expanded polystyrene (EPS), which can be shortened to EIICB. This innovation aims to maximize the potential of producing thermally insulated blocks at full scale, focusing on features such as portability, dry assembly, speedy execution, reusability, inventiveness, high compression, low absorption, and sustainability, which cannot be fully achieved by previous concrete blocks. Compared with conventional concrete blocks, the EIICB reduces concrete material usage and achieves an estimated 73.5% reduction in $CO_2$ emissions. Experimental evaluation showed an average compressive strength of 7.91 MPa, water absorption of 4.74%, and a calculated thermal R-value of 15.05, demonstrating improved thermal efficiency and compliance with structural requirements for non-load-bearing and light structural applications. This work has provided key benefits: transforming waste concrete into valuable materials for affordable and sustainable construction, providing energy-efficient solutions for modern construction, protecting the environment, and aligning with sustainable building practices. The results from this study show the EIICB's ability to promote the construction of low-cost, energy-efficient housing and indicate future opportunities for optimization, full-scale thermal validation, and life-cycle performance assessment.

## 1. Introduction

Rapid urbanization has increased pressure on the environment from the construction industry [1]. The result of this rapid urbanization has increased the amount of plastics entering the waste stream worldwide due to the increased production of plastic, as well as an increase in demolition due to the increase in the amount of

**Data availability statement:** All relevant data are within the manuscript.

**Funding:** The author(s) received no specific funding for this work.

**Competing interests:** The authors declare no conflicts of interest associated with this study.

**Abbreviations:** ASTM, American Society for Testing and Materials; CE, Carbon emission; EIICB, EPS-insulated interlocking concrete block; EPS, Expanded polystyrene; ICBs, Interlocking concrete blocks; IICB, Insulated interlocking concrete block; RAC, Recycled aggregate concrete; NR, Not reported.

construction, and the associated increase in the carbon footprint from manufacturing new materials for construction [2,3]. Expanded polystyrene (EPS) is one of the largest sources of plastic waste globally, with over 99% of EPS waste being disposed of [4,5]. Concrete demolition is also a significant contributor to landfills across the globe [6]. Together, these waste products not only contribute to the carbon emissions associated with construction [7,8] but also increase the challenges associated with managing disposal sites, increase embodied carbon associated with all of the materials used in construction, and slow down the progress toward creating a sustainable circular economy [9,10]. For instance, there are some studies that utilized natural resources to replace toxic chemical compounds [11]. Besides, researchers have begun to prioritize the use of recycled materials in order to offer sustainable environmental solutions for organic pollutants, conserve natural resources, and extend the useful life of materials across the engineering, architectural, and environmental sectors [12,13].

The construction industry is also facing increasing pressure from the government to decrease its carbon footprint while providing a quick, low-cost, and energy-efficient method for constructing buildings [14–16]. Mining areas are particularly unique in this respect because they have extremely challenging weather patterns, highly mobile workforces, and a cyclical pattern of buildings [17–19]. Consequently, the construction of these buildings is subject to a high level of material waste and financial loss. The traditional construction method in these mining areas may not have sufficient structural integrity or thermal efficiency, while the modern methods of construction are structurally efficient, but resource-intensive and high in carbon emissions, creating an urgent need for construction systems that have characteristics of high thermal efficiency, sturdy structure, environmental friendliness, rapid deployment capability, and reusability [20].

The use of interlocking concrete blocks (ICBs) is gaining popularity as a potential alternative to traditional concrete block systems because they can be assembled without the use of mortar, require less labor during construction, and may be reused [21–27]. Most ICBs lack sufficient thermal insulation for many climates because they do not use complete interlocking geometry and require a certain amount of mortar for assembly. Expanded polystyrene (EPS) is a very good thermal insulation product because of its relatively low density, low thermal conductivity, and cost/performance ratio [28–31]. Previous studies have shown that EPS based on a specific combination of materials has been shown to greatly reduce thermal conductivity and improve energy efficiency of buildings constructed with EPS [32,33]. Moreover, EPS blocks can be the cheapest among insulated blocks, and they can reduce thermal conductivity by 19.4% and cut $CO_2$ by about 3.38 times compared to conventional concrete blocks [29]. Despite these findings, there have been very few instances where EPS materials have been used in full-scale interlocking block systems. Similarly, the majority of reported advancements in the areas of green technology, sustainable materials, and nano-enhanced composites for thermal and mechanical properties have not been combined into a reusable and insulated ICB that could facilitate quick construction processes [34–37].

In response to these challenges, this study proposes a novel mortaless EPS-insulated interlocking concrete block (EIICB) that incorporates recycled aggregate concrete (RAC) and EPS insulation core within an ICB system. This fusion combines the structural benefit of ICBs and thermal performance of EPS to create a sustainable, disassembled, and reusable building material that responds to practical demands in creating temporary or semi-permanent housing solutions in areas such as mining camps, urban outlying areas, or disaster recovery efforts. Additionally, since this proposed EIICB incorporates recycled aggregate concrete (RAC) and concrete with reduced volume, it can significantly lower the embodied carbon footprint than conventional concrete. Moreover, the EIICB can also help facilitate achieving global goals for sustainable development by reducing the amount of waste going to landfill from construction sites and the reduction of the use of virgin building materials. This study experimentally evaluates the mechanical properties, the water absorption rate, the ability to retain heat, and the environmental impacts of the EIICB. The results show that the proposed EIICB can offer significant advantages, such as improved energy efficiency, reduced carbon emissions, lower construction costs, and compressive resistance over existing interlocking systems and conventional blocks. Last but not least, remaining limitations and opportunities for future research related to durability, fire safety, long-term thermal behavior, and life-cycle assessment are discussed.

## 2. Experimental design

### 2.1. EIICB constituents

Expanded polystyrene (EPS), as an insulation material, with a density of 3.0 PCF and thicknesses of 1 inch and 1.5 inches, was procured from a local supplier for use in the experiment. Course and fine aggregates were sourced from the demolition wastes in the school compound to reap maximum environmental and economic benefits. Portland cement and gypsum (for capping) were obtained from the stock room at the lab, while tap water was readily available at the institution. Aggregates preparation procedures involved collecting and selecting demolition concrete waste, hand-crushing aggregates into smaller sizes using a hammer, washing to remove surface contaminants, and sieving aggregates using a set of sieves. Fine aggregates comprised 0.075 mm to 4.75 mm, while coarse aggregates were 6–10 mm. Studies indicate that thermally insulated recycled aggregate concrete (RAC) improves thermal insulation properties up to 76.5% [38], offering adequate mechanical properties and significant sustainability advantages [39–41], and substantially raises the shrinkage gradient by 84% to 102% [42].

### 2.2. Concrete mix design

The experiments were performed in the concrete laboratory. The materials were batched by weight and mixed manually. Several trials of different mix ratios of coarse and fine aggregates, cement, water, and EPS were prepared to determine an optimal mix design for the block. After testing, adjustments to the mix proportions were carried out for each trial. Certain factors, such as strength, durability, and workability, were quantitatively considered in determining the most suitable mix for concrete block production. After numerous trials of other mix ratios, specimens with a mix ratio of 1:2:4:4 (water: cement: coarse aggregates: fine aggregates) were opted for.

### 2.3. Mold design

Due to the configuration of the interlocking concrete block, there was no standard mold available. The mold features four hasps at the back and four hinges at the front, allowing it to be easily removed after being turned over, without collapsing the fresh block. To create a standard block that is portable for construction workers and can withstand greater pressure, the following components were designed as shown in Fig 1:

• Front piece: shaped to form the block's projection.

• Back piece: Used to form the block's groove; bottom and half-bottom blocks can be created by flipping the front part.

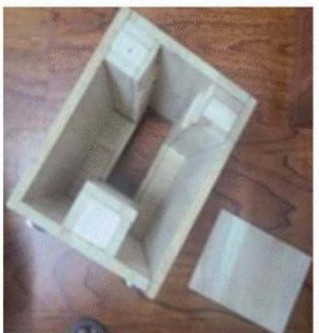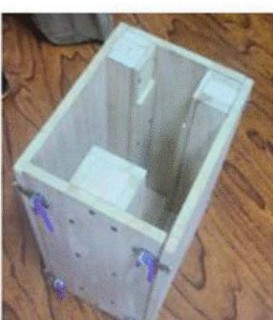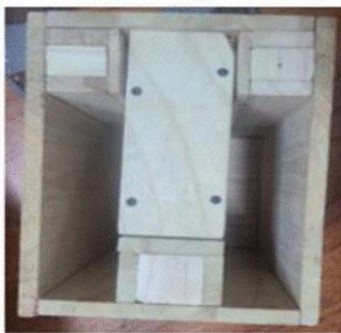

**Fig 1. Timber mold used in specimen preparation shown in multiple views.**

- Bottom base: used to stabilize the mold, preventing the block from slipping during compaction.

- Top piece: Used to create a dent to allow vertical joints to interlock.

The mold dimensions were as follows: the overall length of 46 cm, the total thickness of 20 cm, and the total height of 26 cm. The opposite face of the block was 8 cm thick to allow full interlocking. The side projections were 6 cm thick, and the central interlocking projection was 6 cm in height. For equal pressure distribution during the compression test, the top and bottom blocks were prepared. This was achieved through the insertion of an oiled sheet across the mold during casting to divide the block into two parts. The plan and elevations of the blocks are shown in Fig 2.

## 2.4. The molding procedure

With the concrete mixture prepared, the molding procedure involves the following steps:

- Oil was used to coat all internal surfaces of the mold.

- The wooden mold was positioned on its base and secured using the hasps and locking rods.

- Three pieces of EPS boards were measured and cut as shown in Fig 3(a).

- The designed concrete mixture was prepared.

- Pre-measured and molded concrete spacers were put at the bottom of the mold, and concrete was poured to maintain a 2 cm clearance.

- Each EPS piece was placed centrally within its compartment, maintaining a uniform 2 cm clearance on all sides as shown in the cross-sectional view in Fig 3(b).

- The concrete mix was placed into the mold in three layers.

- Every layer was compacted and stamped 25 times with sufficient force using a steel tamping rod to ensure that no voids were present throughout the entire block, while maintaining the positions of the EPS boards.

- The top insert was placed and pressed to achieve the desired surface and form interlocking dent.

- The mold was opened carefully by releasing the hasps, and the fresh block was demolded by turning the assembly over.

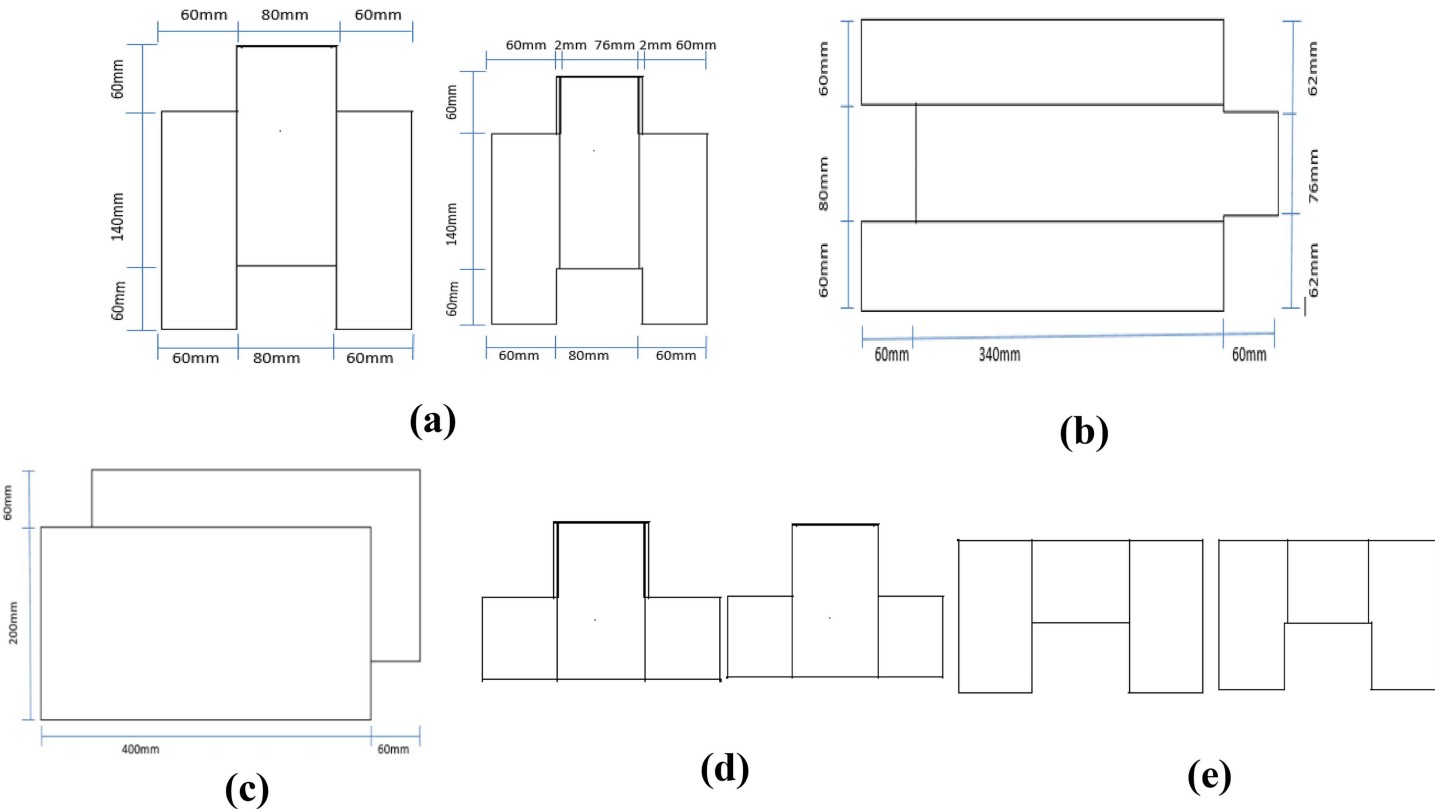

**Fig 2. Plan and elevations of the blocks:** (a) end elevations of the standard block, (b) plan view, (c) front elevation of the standard block, (d) end elevations of the bottom block, (e) end elevations of the top block.

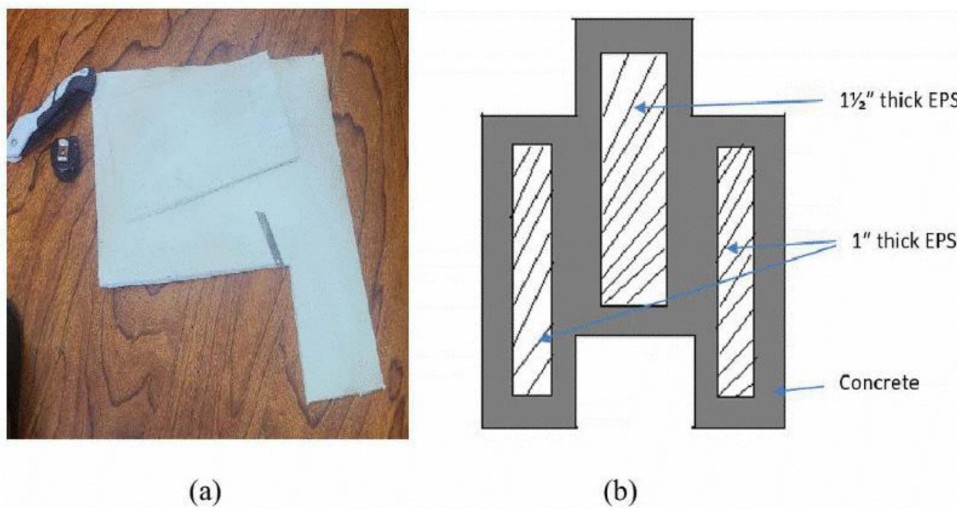

**Fig 3. Components and cross-sectional details of the EIICB system.** (a) EPS insulation board used in the EIICB system, (b) Cross-section of the block across the length.

## 2.5. Curing and capping processes

- All types of blocks were first cured under moist curing conditions for 7 days. (Wet burlap was placed over the surfaces of the blocks to protect against drying out during curing)

- All specimens were submerged in a water tank (water curing tank) for an additional 21 days of water curing (followed by ASTM specifications for masonry units).

After 28 days of curing, all blocks were removed from the water tank, and surfaces were dried in preparation for testing. The specimens were brushed on the surface to remove any residue, and thick gypsum plaster was prepared by mixing it with water. Subsequently, oil was applied on the surface of the steel platens to prevent the plaster from sticking, and a thin layer of gypsum plaster was applied on the steel caps. The block was then carefully placed on the capping plates and pressed gently to form a thin and uniform cap. After the first cap was set for a few minutes, the process was repeated for the opposite sides of the specimens. Finally, the caps were cured overnight before proceeding with the compression test.

## 2.6. Compressive strength testing

An automatic compression-testing machine was used to perform the compressive strength tests, following ASTM C140 and C39 standards [43,44] in terms of load rate application, measurement accuracy, flatness, and compression platen hardness. The load was applied regularly at a rate of 0.3 KN/s until the failure of the specimen. To calculate the compressive strength, the failure load was recorded and divided by the net area of the block, which is normal to the load. Individual blocks were tested first, as illustrated in Fig 4, and the results are calculated using Equation (1). Then, the wallet tests

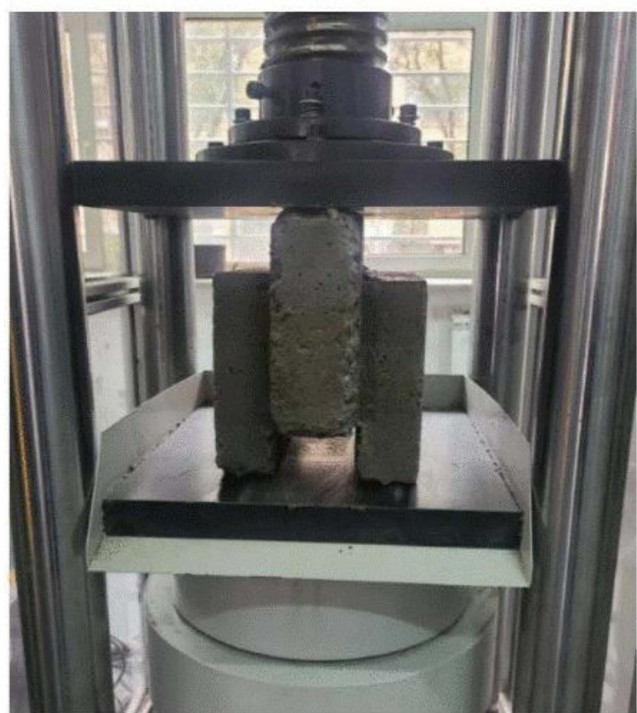

**Fig 4. Standard insulated interlocking concrete block in the compression machine.**

involved arranging a bottom, standard, and top block between two steel plates, aligned on the loading axis as shown in Fig 5, and gradually increasing the load until the wallet failed. Five specimens were tested per case, and compressive strength was calculated based on the net load-bearing area.

$$\text{Compressive strength (MPa)} = \frac{Load\ (KN)}{Surface\ Area\ (mm^2)} \times 1000$$

(1)

## 2.7. Water absorption testing

Water absorption tests were carried out by completely soaking the samples in water for 24 hours. After this period, the samples were removed, and excess water was allowed to drain. The block mass was recorded at this point. The samples were then dried to a constant mass in a dryer circulating hot air, and their mass was recorded once again. The test was performed on five samples. Water absorption was calculated using Equation (2).

$$W_a = \frac{M_w - M_d}{M_d} \times 100$$

(2)

Where, $W_a$ is water absorption; $M_w$ is the mass of the sample saturated with water to a constant mass; $M_d$ is the mass of the sample, which is dried to a constant mass.

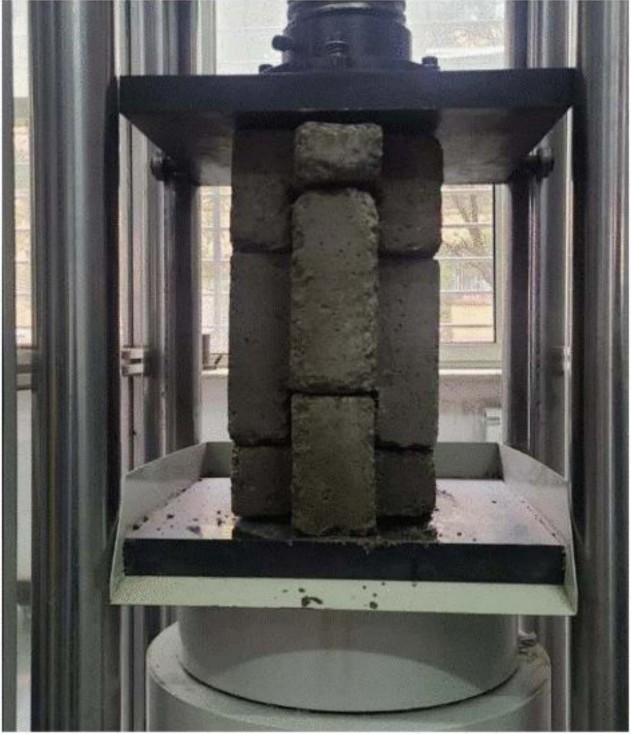

**Fig 5. Three types of IICBs in the compression machine after loading.**

## 3. Results and analysis

### 3.1. Physical characteristics of EIICB

The EIICB unit's method of manufacture resulted in units of consistent geometry (in which the units contained EPS inserts completely encapsulated in concrete with a uniform concrete cover surrounding the EPS). The EIICB was visually inspected for the bond between the EPS insert and concrete, and no voids were found at the surface. The average oven-dry density for all of the blocks produced was between 1,806−1,844 kg/m³, which is less than other conventional concrete products because of the presence of EPS inserts and a reduced concrete volume. Previous studies have indicated that constructing a block using the combination of EPS and RAC will decrease the density of the block while increasing the thermal resistance of the block, and no dimensions were outside the 2-millimeter limit. Therefore, the quality of the EIICB blocks fabricated using this method is satisfactory in terms of quality control during curing and casting.

### 3.2. Mechanical properties of EIICB

**3.2.1. Compressive strength of individual blocks.** The average compressive strength of the block in isolation was 7.91 MPa, with values ranging from 7.73 MPa to 8.07 MPa as summarized in Table 1. The outcomes for individual blocks are presented in a bar graph, as depicted in Fig 6.

Table 1. Compressive strength results of standard block specimens.

| Block number | Load at failure (kN) | Loaded surface area (mm²) | Compressive strength (MPa) | Average compressive strength (kN) | Standard deviation |
|---|---|---|---|---|---|
| B1 | 258.4 | 32000 | 8.07 | 7.91 | 0.11 |
| B2 | 247.5 | | 7.73 | | |
| B3 | 254.7 | | 7.95 | | |
| B4 | 251.9 | | 7.87 | | |
| B5 | 254.4 | | 7.95 | | |

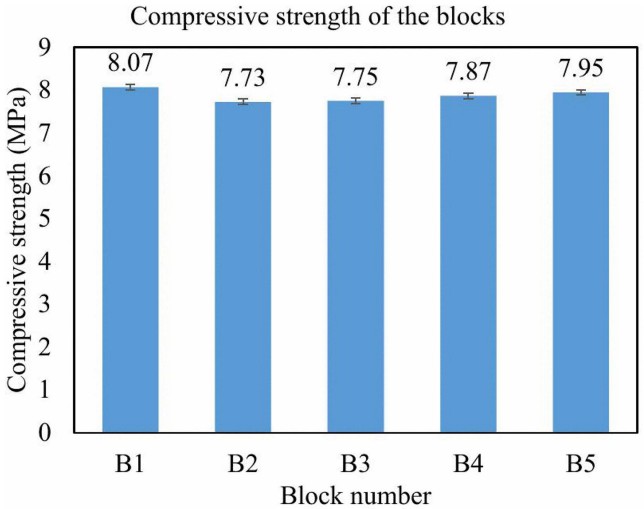

Fig 6. Compressive strength graph for each block.

These values exceed the requirements for non-load-bearing applications as well as light structural applications, e.g., partition walls and low-rise buildings. The variability in strength was likely due to variability in recycled aggregate distribution as well as differences in manual compaction; however, the overall performance of the EIICB met the performance standards for interlocking masonry systems. When compared to a typical hollow block, the EIICB demonstrated superior strength owing to better interlocking geometry and reduced void space.

**3.2.2. Failure mode.** The most common failure mechanism for EIICB blocks was vertical cracking originating from the planes where the shear stress was maximum and moving toward the loading platens. The core of the EPS material was intact at the time of failure, indicating that the transfer of load was a result of the RAC shell surrounding the core. There was no observable delamination between EPS and concrete. The failure mode for isolated blocks is shown in Fig 7.

### 3.3. Wallette compressive testing

Testing of wallette assemblies made of top, standard, and bottom blocks was used to simulate a wall system under normal service conditions. The ultimate loads that were tested and the compressive strength values calculated are presented in Table 2. When tested as part of a wallette, the average strength reduced to 6.40 MPa, with a range of 5.61 MPa to 6.77 MPa. The outcomes for wallets are presented in a bar graph, as depicted in Fig 8.

### 3.4. Water absorption

The load/displacement characteristics were such that the stiffness of the assemblies decreased gradually after the peak load was reached. As anticipated, these loads and stiffness characteristics were similar to those of interlocking masonry systems, where mechanical interlock and friction are primary means by which loads are transferred from one unit to another. The strength values for wallette assemblies were lower than those of individual blocks, but results indicate that there is an opportunity to use EIICB units in non-load-bearing and moderate-load applications.

The average water absorption exhibited by the EIICB units was 4.74%, which is substantially lower than that found in conventional concrete masonry units, which are generally in the range of 6–12%. This was primarily due to the presence of the internal EPS core that substitutes for some of the concrete material and therefore decreases the total amount

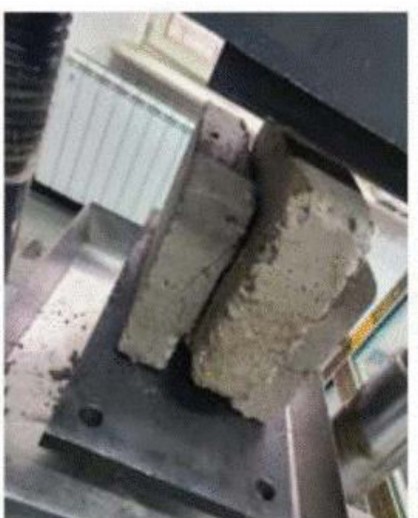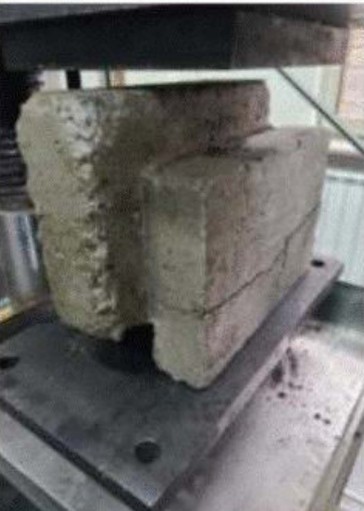

**Fig 7. Failure pattern of the blocks after loading.**

**Table 2. Compressive strength results of wallette specimens.**

| Wallette number | Load at failure (kN) | Loaded surface area (mm²) | Compressive strength (MPa) | Average compressive strength (kN) | Standard deviation |
|---|---|---|---|---|---|
| W1 | 447.6 | 79760 | 5.61 | 6.4 | 0.42 |
| W2 | 540.3 | | 6.77 | | |
| W3 | 509.1 | | 6.38 | | |
| W4 | 534.5 | | 6.70 | | |
| W5 | 521.8 | | 6.54 | | |

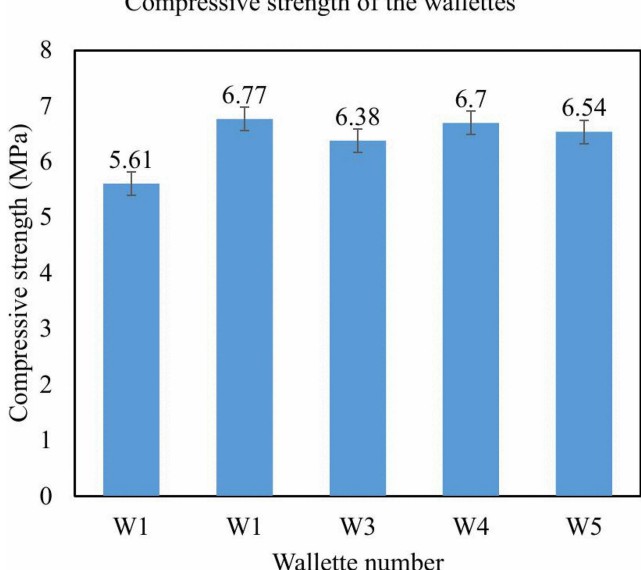

**Fig 8. Compressive strength graph for each wallette.**

of water that can be absorbed into the unit. Low levels of water absorption will enhance long-term durability through decreased potential for moisture-driven deterioration, freeze-thaw fracturing, and surface efflorescence. The results obtained fall within the limits set forth by established international guidelines for quality masonry products. The results are summarized in Table 3 and presented in a bar graph, as depicted in Fig 9.

### 3.5. Thermal performance

Thermal resistance (R-value) of the EIICB was calculated from the known thermal conductivity (k) of EPS using standard specifications for rigid, cellular polystyrene thermal insulation [45]. The composite R-value of the block is estimated to be approximately 15.05 (m²-K)/W, which means it provides better insulation performance than traditional solid concrete blocks. Moreover, EPS accounts for the majority of the thermal resistance, and RAC provides additional thermal resistance by trapping air bubbles in the building material. Although there were no thermal tests performed, a small deviation was expected in the R-value due to small gaps between the EPS boards of which were assumed to be recovered by RAC. The authors recommend future validation of these results by performing thermal test measurements on a full-sized laboratory setup, along with thermal simulations to improve accuracy and pre-compliance with local building codes.

**Table 3. Water absorption test of standard block specimens.**

| Block Number | Mass of saturated block (kg) | Mass of dried block (kg) | Average mass (kg) | Change in mass (kg) | Water absorption (%) | Average water absorption | Standard deviation |
|---|---|---|---|---|---|---|---|
| B1 | 30.83 | 29.50 | 29.17 | 1.33 | 4.51 | 4.74 | 0.26 |
| B2 | 30.41 | 28.97 | | 1.44 | 4.97 | | |
| B3 | 30.62 | 29.30 | | 1.32 | 4.51 | | |
| B4 | 30.37 | 28.91 | | 1.46 | 5.05 | | |
| B5 | 30.55 | 29.19 | | 1.36 | 4.66 | | |

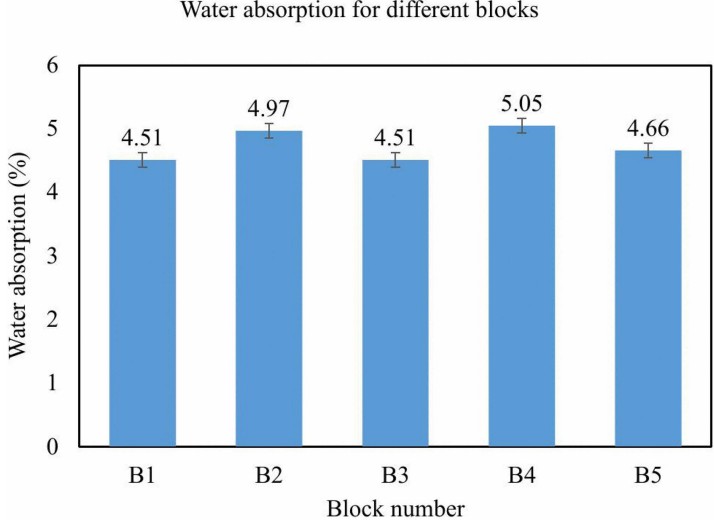

**Fig 9. Water absorption test graph.**

### 3.6. Environmental and material efficiency benefits

The use of the EIICB is estimated to reduce the $CO_2$ emissions by 73.5% compared to traditional manufactured blocks of similar size. There are five reasons this decrease happened so drastically:

- EIICBs used recycled aggregates instead of natural aggregates

- The mortar-free building method reduces cement consumption

- The EIICB system uses a single wall configuration, resulting in approximately a 50% reduction in walling material compared with traditional block construction.

- The internal voids are occupied by the EPS, lowering the overall cement consumption.

- EIICBs may be reused, extending their life and reducing future demolition waste.

In summary, the results have demonstrated the advantages of integrating RAC and EPS and emphasized the contributions a reusable concrete product, such as the EIICB, can make to a circular economy in construction.

## 4. Discussions

The construction industry never stays still as it keeps seeking solutions to problems that arise in a particular context, such as in the case of mining sites with weak ground, extreme climatic conditions, or high humidity. The new EIICB may

be considered historically new and ecologically sound material that is extremely efficient for use in residential construction in these areas. This block is mortar-free, made of recycled concrete, and has the following performance characteristics: average compressive strength of 7.91 MPa, water absorption capacity of 4.74%, and R-value of 15.05. However, a small deviation is expected in the R-value due to small gaps between the EPS boards. Results indicate that the overall mechanical strength, environmental benefits, thermal resistance, and constructive efficiency of the EIICB are distributed evenly across all areas of development. This section discusses how these findings contribute to the understanding of these areas of development, compares them against available literature, and offers practical implications and limitations of the EIICB.

### 4.1. Block geometry and masonry assembly

The EIICB units were manufactured with nominal dimensions of 460 × 200 × 260 mm (length × width × height). The blocks are designed to be assembled using a dry-stacked interlocking system, eliminating the need for conventional mortar joints. Horizontal and vertical bonding between adjacent blocks is achieved through complementary interlocking protrusions and recesses, which provide geometric confinement and ensure proper alignment during construction. The interlocking configuration enables effective load transfer and shear resistance between block layers while maintaining consistent spacing. When assembled, the EIICB system forms a staggered running bond arrangement comparable to a traditional stretcher bond, thereby enhancing structural stability and constructability without reliance on mortar.

### 4.2. Assessment of mechanical strength

The average compressive strength (7.91 MPa) exhibited by the EIICB block indicates that it can fulfill the criteria required for either non-load-bearing applications or specific low-load-bearing applications. Despite the fact that EPS decreases the concrete cross-section available, the strength of this block is equivalent to or exceeds that found in many conventional hollow or interlocking blocks, as compiled by others [31,32,46–48]. Other similar investigations have shown that if RAC is correctly graded and properly proportioned, the structural qualities will be within acceptable limits. When tested as part of a wallette, the average strength reduced to 6.40 MPa. The compression machine stopped automatically once the wallette failed, and in all cases, it was observed that the top or bottom block failed. The reduction in the compressive strength for part of a wallette can be attributed to several factors.

Firstly, the load distribution across interlocking interfaces plays a significant role, as loads in interlocking blocks are distributed across several interfaces formed due to block geometry. The interfaces may create high concentrations of stress locally, especially if they occur at the points in contact or edges of interlock, resulting in lesser load-bearing capacity than for an individual block [49–51]. Secondly, stress transfer and effects from assembly differ from single block tests; in wallettes, stress can be transferred between interconnected blocks, which causes localized failures due to imbalanced pressure distribution resulting from either imperfect interlocking or misalignment during construction [52,53]. Additionally, geometric imperfections, such as slight imperfections in block geometry or misalignment of interlocks, can intensify stress concentrations, resulting in a loss of uniformity in the load distribution across the wallette assembly and a corresponding reduction in strength [54,55]. Furthermore, mortar-free assemblies depend upon interlocking mechanisms for their lateral stability, which can trigger occurrences of minor displacements or slight rotations of interlocks displaced under compressive loading, thereby reducing the overall shear strength and bringing about earlier failure as compared to tests performed on individual blocks [56–58].

The results demonstrate the influence of assembly behavior on the structural performance of mortar-free interlocking systems, which is a benefit for further optimization of interlocking block designs and construction practices expected to improve compressive strength and corresponding reliability of masonry assemblies for practical applications. The results indicated that all blocks had exceeded 3.45 MPa in compressive strength, which is the minimum requirement for non-load-bearing concrete blocks according to ASTM C129 [59].

### 4.3. Water absorption and durability considerations

The water absorption percentages range from approximately 4.5% to 5.05%, with an average of 4.74%. For insulated concrete blocks, water absorption should usually be less than 10% by weight after 24 hours of immersion in water (ASTM C90) [60,61]. The primary reason for the lower water absorption rate of the EIICB is the replacement of such a large portion of concrete volume with an EPS shell. Low water absorption contributes to increased durability because the block is less prone to moisture-induced damage, such as cracking, swelling, and mold infestations. For the coastal areas and mining regions that are prone to floods or humidity, this feature will extend the performance of the building for considerable periods and enhance cost savings on maintenance.

The long-term durability of the EIICB must still be evaluated closely. Although the EPS shell does not absorb water, it may be susceptible to degradation due to extended exposure to high temperatures. Therefore, future tests should be performed to provide an accelerated testing program, a chloride penetration analysis, and a long-term moisture cycling analysis to determine the potential of the EIICB to perform effectively under actual environmental conditions.

### 4.4. Thermal insulation performance

The EIICB offers potential for energy-efficient construction due to its estimated R-value of 15.05. This R-value demonstrates thermal resistance that exceeds that of traditional concrete blocks, which often have very low insulation qualities due to high thermal conductivity. The majority of the EIICB's thermal resistance comes from its EPS envelope; however, RAC's heterogeneous microstructure and entrapped air also provide additional insulation. Research shows that EPS is effective for insulating buildings and has shown significant reductions in structural weight and economical designs [32,62]. The research presented here is consistent with this conclusion and demonstrates that EIICB can be a substantial contributor to maintaining a comfortable indoor temperature, reducing energy loss, and enhancing energy efficiency.

The R-values presented in this research were estimated based on the manufacturer's specifications. The insulation performance of EPS depends on the thickness and density of the EPS shell used. Since the block contains a 3.5-inch EPS core with a density of 3.0 PCF, it will achieve an R-value of 15.05 [45]. The U.S. Department of Energy recommends insulation on exterior walls with R-values of R-13 to R-23 to improve energy efficiency in buildings [63]. Future studies should include standardized thermal testing (guarded hot-box or heat-flow meter) to confirm these results through testing. In addition, thermal bridge analysis, along with computational thermal simulation programs, can assist designers in optimizing their designs.

### 4.5. Environmental and resource efficiency implications

Based on the study of EIICB, $CO_2$ emissions from EIICB are expected to be reduced by 73.5% when compared to those of a normal concrete block. This is an enormous environmental saving that not only comes from the replacement of aggregates (which is done with recycled concrete) but also due to a combination of reduced cement usage from the expanded polystyrene core (EPS) and reusability of the blocks, thereby reducing demolition waste and the lifecycle impacts associated with these blocks. The findings of the study are consistent with the principles of the global circular economy, which emphasize the recovery of materials and the reduction of waste. Other studies have indicated that using recycled concrete aggregates will reduce the embodied carbon of RAC, while EPS provides additional environmental benefits when reused [39–41,64]. The EIICB takes this one step further by combining both benefits into one reusable building unit.

### 4.6. Practical applications and construction benefits

#### 4.6.1. Eco-friendly and resource-efficient building innovations.
More materials used in construction lead to higher $CO_2$ emissions [65]. To cut the growing $CO_2$ emissions related to construction, it is important to have low-emission buildings with lower material consumption. The reuse of building materials and recycled concrete can greatly reduce

emissions associated with new material production, thus supporting the tenets of the circular economy while also reducing the environmental impact of concrete production. Innovative block designs with insulating inserts bring thermal and resource efficiency, as shown by Hanan and Khalid [28]. The block will acquire a high R-value and provide thermal insulation in areas faced with harsh and extreme temperature ranges, which ultimately results in lower energy expenditure in temperature regulation.

**4.6.2. Sustainable and efficient construction practices.** Many mining areas are not privately owned, and when mining ceases, miners often leave behind unused structures. In past decades, bamboo has been widely used for temporary and permanent structures, whereby studies have been conducted to improve its strength by filling concrete in its cavity [66]. The proposed EIICB can be easily reused, offering flexibility for both permanent and temporary structures, as miners may need to relocate when mines are exhausted. By adopting sustainable practices, the environmental footprint can be aligned with broader sustainability goals, such as reducing resource consumption, minimizing waste and cost, and encouraging the use of recycled materials [67]. Additionally, the benefits of faster construction become evident with mortar-free building methods that resemble prefabrication, significantly shortening construction time while reducing labor costs and enhancing overall quality [68]. Research by Ramamurthy and Nambiar [69] reported that labor productivity in constructing walls using mortarless masonry can increase by 3–5 times. The mortar-free structure increases the speed of construction and minimizes labor costs because there is no need to bind mortar. This feature eases the building process, especially in far-flung mining sites where skilled labor and materials are often lacking. Furthermore, mortar-free systems enhance accuracy and structural uniformity, which in turn minimizes material wastage. Economically, dry-stacked concrete masonry construction involving laying units without mortar accelerates the process and lowers masonry installation costs [70]. In the case of traditional methods, one could use a cavity wall with two parallel layers of blocks and a cavity filled with insulation materials to enhance energy efficiency and keep warm for buildings [71,72], which results in an increase of construction materials. Studies showed that the new single-layer blocks, characterized by low thermal conductivity and high thermal inertia, deliver exceptional performance by significantly reducing indoor air temperature compared to alternative solutions [73]. Moreover, compared with a conventional cavity wall of the same R-value of 15 and using similar materials, the proposed EIICB has the advantage of reducing 73.5% CE due to concrete material reduction. Using fewer materials without sacrificing the functionality of the building is a significant advancement for the construction industry. The reuse of materials and reduction in building requirements will greatly lower costs, attracting more investors and helping address housing shortages.

**4.6.3. Efficient transportation.** In contrast to prefabrication, which requires specialized transportation equipment, studies have highlighted the potential of on-site management and standardized design by utilizing innovative forms of precast component factories, which can minimize transportation costs [74]. The portability and ease of assembly of these proposed EPS-IICBs simplify logistics, eliminating the need for special equipment and streamlining transportation and installation.

## 4.7. Comparison with existing block systems

Different types of blocks are locked differently, and this subsection is specific about both directions of locking. To the best of the authors' knowledge, only one available horizontal locking production is found in the literature. The interlocking properties of the proposed EIICB are compared to the contribution of Yılmaz et al. [21], whose aim is to produce a mortar-free block but can only provide horizontal joint locking, leaving the vertical joints open. Although this contribution is commendable in reducing mortar, the proposed EIICB performs better by offering both horizontal and vertical interlocking and being entirely mortar-free, as shown in Fig 10. This innovative EIICB not only improves the structural resistance of the blocks but also simplifies the construction process by reducing labor and material costs associated with mortar application.

Moreover, the insulation properties of EIICB are compared with those studied by previous studies [28,75]. However, while their design uses visible polystyrene insulation on the surface of the block, the proposed EIICB takes a more

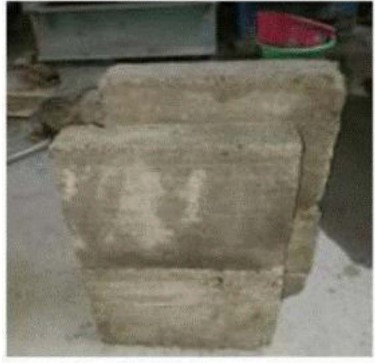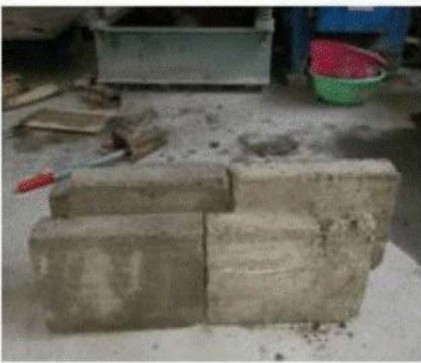

**Fig 10. Interlocking setups of the blocks.**

integrated approach by incorporating insulation directly within the block structure instead of using visible polystyrene insulation on the surface of the block. EIICB's innovative approach enhances the continuity of concrete, leading to improved structural integrity while simultaneously providing effective insulation [76,77]. Moreover, some previous blocks ignored the interlocking capability [28,75], while EPS-IICB's interlocking mechanism enhances its structural resilience and ease of installation. This seamless incorporation of insulation and interlocking features may provide a more effective design, cost savings, and be eco-friendly.

In addition, this invention has advantages compared to relatively similar products proposed in the study of [78], which produced a prototype of a block with insulation at a smaller scale (1:10 ratio) without full-scale data. The practical appraisals were missing in their open design, raising concerns about how feasible that design is, especially considering the special tools needed for its heavyweight design when produced at full scale. Unlike their prototype, this study successfully developed a fully functional portable EIICB that minimizes logistical problems and reduces the cost of using expensive reinforcing bars. Furthermore, this study maintains soundness in incorporating practical compressive and absorption tests and a rigorous study of the thermal efficiency of the EPS material utilized, to provide clear evidence for the superiority of insulation effectiveness. This research not only addresses the practical challenges of portability and cost but also delivers reliable data demonstrating superior performance in thermal insulation.

The proposed EIICB is also compared with existing interlocking and insulated block systems, which are summarized in Table 4. It can be found that the EIICB not only exceeds previously proposed blocks in terms of its functionality but also paves the way for sustainability, services, and energy efficiency. Interlocking technology with integrated thermal insulation will serve as a solution of fusion for modern demands regarding eco-friendly, cost-effective, and resilient construction materials.

In general, expanding demand for sustainable construction materials requires more innovative solutions to minimize the carbon footprint while achieving structural soundness. Traditional concrete blocks, which are being extensively used, have led to the depletion of resources and the generation of waste. In this regard, the reusable EIICB that uses recycled aggregates presents a promising alternative. This proposed block provides a good chance of recycling materials, thus reducing landfill waste, while the presence of EPS provides for enhanced thermal insulation, thereby addressing energy efficiency in buildings. Its interlocking nature facilitates easy assembly, disassembly, and reuse, advancing the circular economy within construction. Lastly, in mining areas, geographical limitations or inadequate infrastructure may hinder the sourcing and conveyance of other sustainable materials such as prefabrications. That is to say, though these sustainable materials aim to lower the environmental impact, in mining areas, they will still add to carbon emissions and entail resource depletion since their source could be long distances away. In contrast, the production of the EIICB is so cheap,

**Table 4. Comparative analysis of existing interlocking and insulated block systems.**

| Name | Insulation | Inter-locking | Recycled materials | Mortar-free | Compressive strength reported | Environmental Assessment | Key limitations | Reference |
|---|---|---|---|---|---|---|---|---|
| Conceptual design of interlocking blocks | Yes | Partial | Partial | Partial | Yes (cube only) | No | No block test No LCA | [79] |
| Interlocking and insulated construction block | Yes (unspecified) | Yes | Not reported (NR) | NR | NR | No | No experimental validation | [80] |
| Insulated interlocking superblocks | Yes | Partial | NR | Partial | NR | No | No materials or cost details | [81] |
| Building interlocking blocks and using insulating mortar | Yes | Partial | NR | Partial | NR | No | Sustainability concern (coal ash) | [82] |
| Lightweight interlocking cement block | Cavity | Semi | No | No | Yes | No | Low strength:mortar required | [24] |
| Two-piece interlocking block | Cavity | Yes | NR | Partial | NR | No | No testing or LCA | [25] |
| Building envelope prefabricated with 3D printing technology | Yes | Yes | NR | Yes | NR | No | Prototype scale only | [78] |
| EIICB | Yes | Yes | Yes | Yes | Yes (block and wallette) | Yes | Identified and discussed in Discussion section | This study |

and the transportation is easy for a mining community.. Moreover, when miners relocate, the blocks can be disassembled and reused to build new structures, offering a revolutionary approach to sustainability in the construction sector.

### 4.8. Limitations and future work

This study acknowledges the limitations arising from the restricted scope of the literature review related to interlocking concrete blocks insulated with EPS. Several studies have determined individual aspects of interlocking concrete technology and EPS insulation independently, but comparatively few studies have specifically analyzed their integration and interaction. Moreover, since some of the existing literature may not account for the latest developments or potential innovations in this niche area, this could further reduce the data collection range and encumber the applicability of the findings. The few studies available may differ widely regarding methodology, objectives, and contexts, thus making it very difficult to establish direct comparisons or to formulate uniform conclusions that can be easily applied to other contexts. While the EIICB gives a good show for mechanical properties, thermal, and environmental performance, limitations exist that require acknowledgement so that practical implementations can be realized and provide a pathway towards future developments.

**4.8.1. Fire safety and heat conductivity.** The fire performance of EPS insulation systems must be addressed in the context of fire safety. The present research did not consider the fire performance of the EIICB. Future research should include standardized fire testing and explore the potential of utilizing fire-resistant coatings, creating mineral-based barriers, and considering the modification of block geometry to achieve higher fire performance.

**4.8.2. The source of material and variation.** The recycled aggregates used in this study were from managed waste from the campus demolition site. In an actual construction site, the type, makeup, and amount of contamination associated with the demolition waste can be different. This sort of variation may affect both the mechanical strength and perseverance of the composite as well as the moisture resistance/resistance characteristics. Therefore, it is recommended that further investigations into a broader base of waste need to be undertaken, as well as the development of standardized pre-processing methods to provide uniformity of the properties of materials used to create the EIICB.

**4.8.3. The effect of ultraviolet light on EPS and future use.** Under exposure to ultraviolet (UV) light, certain temperatures, and moisture cycles of an extended duration, expanded polystyrene (EPS) is susceptible to degradation.

Although the EPS component is fully encapsulated by the concrete shell, as shown by the EIICB design, there may be degradation or insulation failure due to micro-cracking or long-term exposure to the environment. Therefore, future investigations are needed in order to design accelerated age testing and cyclic temperature loading. Likewise, it is required to establish historical measurements in terms of the thermal properties of the materials.

**4.8.4. Verification of thermal resistance.** The thermal resistance (R-value) reported in this study was calculated based on established thermal conductivity values in the manufacturer's standard specifications. A full-sized laboratory validation (e.g., through guarded hotbox or heat-flow meter methods) is necessary for the thermal performance verification under actual operating conditions.

**4.8.5. Structural/durability performance.** The compressive performance evaluation of the EIICB provides an incomplete picture of the structural performance of the EIICB. Therefore, additional structural tests need to be conducted as follows: shear strength, flexural resistance, seismic behavior, and interlock slip resistance. In addition to structural performance tests, durability tests such as chloride penetration testing, freeze-thaw cycling, and carbonation depth measurement need to be undertaken.

**4.8.6. Assessing sustainability and economic impact.** Although the benefits of the EIICB are represented in terms of the reduction of greenhouse gases via decreased $CO_2$ emissions, an in-depth and robust life cycle assessment is required. The life cycle assessment can examine the manufacturing cycle of EIICB's raw materials, as well as the end-of-life scenarios, and provide valuable insight into the sustainability impacts of the EIICB. Additionally, an economic viability analysis, including both cost-benefit analyses and capital cost assessments, will provide a basis for comparison between EIICB and conventional masonry systems.

**4.8.7. Production capacity and viability.** The equipment used to manufacture the EIICB system was manufactured with a wooden hand-built mold, and they utilized a process that limited precision and ultimately output capacity. Large volume production may require the use of steel production molds, mechanized casting systems, or prefabricated plastic molds. Further investigation into the various industrial manufacturing processes with respect to tolerances, delivery controls, and quality will be beneficial.

**4.8.8. Implementing the EIICB system and adhering to construction codes and related compliance regulations.** The EIICB offers advantages with respect to temporary construction, affordable housing, and in those regions where climate concern exists. However, its full incorporation into mainstream construction will require being in accordance with the National and International Building Codes. Further effort needs to be put into comparing EIICB's ability against the relevant regulatory standards for thermal insulation, structural load, as well as moisture and heat resistance, to help validate and demonstrate its application in the actual marketplace.

## 5. Conclusion

An innovatively mortarless EPS-insulated interlocking concrete block (EIICB) that incorporates recycled aggregate concrete (RAC) and expanded polystyrene (EPS) insulation core within an ICB system, is designed and proposed in this study to fulfill the requirement of mortar-free insulated interlocking concrete blocks, which could be adopted as a promising solution for reusable, thermal efficient, and sustainable construction. The research aims to address the growing number of environmental challenges that we face today due to the demolition of buildings, the production of plastics, and the problems associated with using non-renewable resources in building construction.

With a calculated average **compressive strength of 7.91 MPa**, the EIICB satisfies or exceeds the standard structural provisions allowing its use in residential and light commercial structures. Such compressive strength renders these blocks' load-bearing wall systems suitable and durable in structural applications. In addition, the blocks showed a **water absorption of 4.74%**, indicating good resistance against moisture penetration. This low absorption rate makes the blocks more resilient against areas where moisture exposure can occur, minimizing issues of water damage as time goes on, and hence, this block can be used under humid or wet climates based on its long-term performance. The estimated **R-value of**

**15.05** highlights the block's insulation potential over that of solid and hollow traditional blocks. Furthermore, the proposed EIICB offers the benefit of **decreasing CE by 73.5%** due to the use of recycled aggregates, reduction in cement demand, and overall concrete material production.

The results further confirm the potential of mortarless EIICB as a compelling option for traditional building materials. Compared with previous innovative IICB products, the proposed EIICB not only exceeds previously proposed blocks in terms of its functionality but also paves the way for sustainability, services, and energy efficiency. The proposed EIICB could provide a low-cost and energy-efficient modern construction component in sustainable building practices. This research has delivered significant contributions by enabling the conversion of waste concrete into high-value construction resources, promoting cost-effective and environmentally sustainable building solutions, enhancing energy efficiency in contemporary construction practices, mitigating ecological impacts, and advancing principles aligned with sustainable development goals in the built environment. Despite these promising outcomes, future studies are required to systematically analyze significant gaps in their fire resistance, long-term strength and stability, standardized thermal performance testing, and the capability of large-scale production of these blocks. Addressing these gaps will allow for the incorporation of the EIICB into building codes and industry practice. Overall, the findings indicate that EIICB has the potential to provide an important step in furthering the development of low-carbon, resource-efficient, and energy-conscious building systems. The integration of both recycled content and insulation into a reusable block system provides an excellent avenue for advancing sustainable building systems for both developed and underdeveloped regions.

## Author contributions

**Conceptualization:** Hui Gao.

**Data curation:** Hui Gao, Moses Mwenda Karani, Zhongwei Zhao.

**Formal analysis:** Hui Gao.

**Investigation:** Li Zuo.

**Methodology:** Hui Gao.

**Resources:** Hui Gao, Moses Mwenda Karani, Li Zuo.

**Supervision:** Zhongwei Zhao.

**Validation:** Hui Gao, Zhongwei Zhao.

**Visualization:** Moses Mwenda Karani, Li Zuo.

**Writing – original draft:** Hui Gao, Moses Mwenda Karani.

**Writing – review & editing:** Hui Gao.

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
