## [Decision Letter · Decision Letter 0]

17 Nov 2025

Dear Dr. Karani,

We look forward to receiving your revised manuscript.

Kind regards,

Pratheep K. Annamalai

Academic Editor

PLOS ONE

Journal Requirements:

Additional Editor Comments:

1. Please broaden the Introduction for a multidisciplinary audience or readership of PLOS one journal.

The current introduction is narrowly focused on mining regions and technical aspects of interlocking blocks. Please revise to educate a wider audience about the global challenge of recycling plastic (EPS) and demolition waste in construction.

Authors may please begin with the environmental context, highlighting the scale of plastic and concrete waste, their impact on landfills and carbon emissions, and the role of circular economy principles, and then clearly explain why integrating EPS and recycled aggregates into reusable blocks is significant for sustainability, energy efficiency, and resource conservation across multiple disciplines (civil engineering, environmental science, architecture, policy).

2. In Experimental Methods section, a clear description needed, for a formal scientific narrative. Example, the current methods read like a procedural manual with command like sentences (“Position the mold,” “Prepare the mixture”). So authors may please revise to passive voice and journal style, e.g.,

“Expanded polystyrene boards were procured… Aggregates were collected from demolition waste, crushed, and sieved… Concrete mixtures were prepared and molded in timber forms…”, please ensure consistency in describing steps (procurement, preparation, moulding, curing) and include precise technical details (dimensions, curing conditions, standards followed). Also importantly avoid conversational tone; adopt clear, concise, and replicable language suitable for peer-reviewed journals.

3. In R&D section: Please strengthen the cross correlation and strengthen the arguments or interpretations.

Please expand beyond reporting results to address practical implications and limitations:

If possible, please validate thermal performance experimentally or via simulation under real conditions, discuss fire safety and durability of EPS insulation and propose mitigation strategies, include life cycle assessment (LCA) and economic feasibility to quantify sustainability benefits and discuss or address scalability, compliance with building codes, and potential adoption challenges.

Auhtors may please consider comparing EIICB performance with conventional and alternative systems in terms of cost, energy efficiency, and environmental impact, if these data are available in other literature.

4. In end of R&D or conclusion section, please acknowledge limitations in sourcing demolition waste and EPS degradation over time, and suggest future research directions.

Reviewers' comments:

Reviewer's Responses to Questions

**Comments to the Author**

1. Is the manuscript technically sound, and do the data support the conclusions?

Reviewer #1: Yes

Reviewer #2: Partly

2. Has the statistical analysis been performed appropriately and rigorously?

Reviewer #1: Yes

Reviewer #2: Yes

3. Have the authors made all data underlying the findings in their manuscript fully available?

Reviewer #1: Yes

Reviewer #2: No

4. Is the manuscript presented in an intelligible fashion and written in standard English?

Reviewer #1: No

Reviewer #2: No

Reviewer #1: In this work, an enveloped insulated interlocking concrete block (IICB) made of recycled aggregates concrete (RAC) and expanded polystyrene (EPS), which can be shortened to EIICB, was investigated. This manuscript will be considered for acceptable after revised based on the following comments:

1. Abstract must be enriched via valuable results which pave the way for understanding the audiences.

2. Please start the abstract by a short introduction of the current problem(s) and the solution, based on the current study, in one or two lines.

3. What the main significance of paper in comparison is of relates published works?

4. Written is very week. In its current state, the level of English throughout the manuscript needs language polishing. Please check the manuscript and refine the language carefully.

5. The introduction section is very short and poorly described. It doesn't present the reference to the manuscript scope. In the introduction section, authors should make an in-depth literature review concerning the use of new technologies such as green nanotechnology to improve human life, construction and solve environmental and economic problems as well as storage and production of clean energy via simple and economical methods, and sustainability. Introduction has deficiency citation to valuable works published before. The following references are recommended to be cited, to improve the introduction section:

Composites Part B 167 (2019) 643–653; Journal of Environmental Management 350 (2024) 119545; Scientific Reports, (2023) 13:19042, 10.1038/s41598-023-46536-8; Scientific Reports, (2024) 14:10914, 10.1038/s41598-024-61688-x;

6. And the structure of the manuscript might need a major adjusting for a better understanding.

7. The novelty of this study should be inserted in the text clearly.

8. Results and discussion: - To increase the scientific value of the manuscript Authors should consider extension of the all results section with comparison of obtained results with the results described in previous publications.

9. How does your paper contribute to the advancement of knowledge?

10. What are the gap areas and the new contribution in the paper?

11. The authors should prepare all figures with better resolution.

12. The authors should prepare all tables with better quality.

13. This work should be compared with the other work in Table form.

14. Indeed, there are impressive amount of results. However, the conclusions section needs to improve with selected and highlighted main findings.

15. The discussion is poor and should be improved.

16. Where is the practical application of this manuscript? It must be added.

17. The abstract needs to be revised. Please further organize your views and look to the future.

18. The language of the manuscript is very pitiful, due to which several times it is hard to understand exactly what the authors want to tell. Hence, the language must be improved by a professional service.

19. The authors are recommended to be cited important references related to using environmentally friendly technologies to solve environmental problems, in the field of health, wellness, treatment, and improving people's lives, environmental remediation, and achieve sustainable development, such as: Results in Engineering 23 (2024) 102399; Results in Engineering 24 (2024) 103281; Results in Engineering 25 (2025) 103758

Reviewer #2: This paper examines a novel EPS-insulated, mortar-free interlocking block using recycled aggregates. The work presents initial mechanical and physical test results; however, important methodological details and additional evaluations are needed to support the conclusions.

1. The word/property of insulation is very focused in the paper it would be great if this could be added in the title

2. There is no paper from 2025 which seems to be concerning for a submission in the last quarter of the year

3. Figure no 1 may be labelled properly with a comprehensive caption

4. Clarify the thermal R-value: State clearly whether the R-value is measured or calculated. If measured, provide full test method, standard apparatus, and raw data. If calculated, provide formulas, thermal conductivity values, assumptions, and uncertainty.

5. Provide details/references for the CO₂ reduction claim

6. Fully characterize the recycled aggregates and other materials for reproducibility

7. The mix design may be written in standard engineering format

8. The authors are encouraged to describe and standardize the compaction method for getting reproducible details (number of tamps, applied energy, or vibration method).

9. How to 2 cm clearance on all sides was maintained?

10. How? Cure the blocks for one week and then submerge for 21 days

11. Figure 3a may be appropriately labelled

12. The authors are encouraged to add references about the quantitative insulating properties of EPS and theirs effect in the current study

13. The number of test for compression and its technical discussion is less.

14. Future recommendation as per the article idea may be added at relevant location

15. What was applied on the surfaces of block for uniform stresses?

16. Regarding the dimensions of the block? What is the solution of joints and bonding in layers? What type of masonry bond will it produce?

17. In table 4 revise/rephrase detailed numerical result

18. Rephrase heading 4.2.1

19. In heading 4.2.5. Low absorption this may be changed to low water absorption

**Do you want your identity to be public for this peer review?** For information about this choice, including consent withdrawal, please see our Privacy Policy

Reviewer #1: No

Reviewer #2: No

---

## [Author Response · Author response to Decision Letter 1]

30 Dec 2025

Response to Editor and Reviewers

The authors thank the Academic Editor and the Reviewers for their constructive comments and valuable suggestions. The authors have revised the manuscript thoroughly, addressed all comments, and improved the overall scientific quality, clarity, and organization. Below is a point-by-point response. All changes have been incorporated into the revised manuscript.

EDITOR COMMENTS

Editor Comment 1

Please broaden the Introduction for a multidisciplinary audience or readership of PLOS one journal.

The current introduction is narrowly focused on mining regions and technical aspects of interlocking blocks. Please revise to educate a wider audience about the global challenge of recycling plastic (EPS) and demolition waste in construction.

Authors may please begin with the environmental context, highlighting the scale of plastic and concrete waste, their impact on landfills and carbon emissions, and the role of circular economy principles, and then clearly explain why integrating EPS and recycled aggregates into reusable blocks is significant for sustainability, energy efficiency, and resource conservation across multiple disciplines (civil engineering, environmental science, architecture, policy).

Response:

Thank you very much for the kind suggestions. The Introduction has been substantially rewritten to provide a broader multidisciplinary context. We now begin with the global challenge of plastic (EPS) waste and concrete demolition waste, their environmental impacts, landfill burdens, and associated carbon emissions. Principles of circular economy, resource conservation, and sustainability across civil engineering, environmental science, materials engineering, architecture, and policy have been added. We also describe why integrating EPS and recycled aggregates into reusable blocks is important for global sustainability.

Changes made in manuscript: Revised Introduction: Lines 50–94 (first three paragraphs fully rewritten and expanded).

Editor Comment 2

In Experimental Methods section, a clear description needed, for a formal scientific narrative. Example, the current methods read like a procedural manual with command like sentences (“Position the mold,” “Prepare the mixture”). So authors may please revise to passive voice and journal style, e.g.,

“Expanded polystyrene boards were procured… Aggregates were collected from demolition waste, crushed, and sieved… Concrete mixtures were prepared and molded in timber forms…”, please ensure consistency in describing steps (procurement, preparation, moulding, curing) and include precise technical details (dimensions, curing conditions, standards followed). Also importantly avoid conversational tone; adopt clear, concise, and replicable language suitable for peer-reviewed journals.

Response:

Thank you very much for the kind suggestions. The entire Experimental Methods section has been rewritten in passive, formal scientific style. Imperative/command-type sentences have been removed. The authors added details regarding procurement, preparation, mold dimensions, curing environment, standards used, and procedural consistency.

Changes made in manuscript: Lines 160-263.

Editor Comment 3

In R&D section: Please strengthen the cross correlation and strengthen the arguments or interpretations.

Please expand beyond reporting results to address practical implications and limitations:

If possible, please validate thermal performance experimentally or via simulation under real conditions, discuss fire safety and durability of EPS insulation and propose mitigation strategies, include life cycle assessment (LCA) and economic feasibility to quantify sustainability benefits and discuss or address scalability, compliance with building codes, and potential adoption challenges.

Auhtors may please consider comparing EIICB performance with conventional and alternative systems in terms of cost, energy efficiency, and environmental impact, if these data are available in other literature.

Response:

Thank you very much for the kind suggestions. The R&D section has been significantly expanded. The revisions include:

Comparative discussion with conventional and alternative insulated/ICB systems

Emphasis on durability, fire safety considerations for EPS

Initial LCA considerations for concrete material reduction and CO₂ savings

Economic/feasibility discussion and scalability

Practical applications and compliance with building codes

Clarified insulation performance

Limitations of EPS degradation over time

Future research directions

Changes made in manuscript: Discussion Section: Lines 731-736, & 753-785 (new subsections added).

Editor Comment 4

In end of R&D or conclusion section, please acknowledge limitations in sourcing demolition waste and EPS degradation over time, and suggest future research directions.

Response:

Thank you very much for the kind suggestions. A new Limitations and Future Research subsection has been added. It includes:

Challenges in sourcing uniform demolition waste

EPS longevity and degradation issues

Need for more comprehensive LCA

Need for fire performance testing

Need for larger-scale structural and thermal validation

Changes made in manuscript: Lines 737-752 in the marked-up revised manuscript,

REVIEWER #1 COMMENTS

1. Abstract must be enriched via valuable results which pave the way for understanding the audiences.

Response:

Thank you very much for the kind suggestions. The Abstract has been improved to include clearer numerical results, contribution highlights, and significance.

Changes made in manuscript: Lines 19-23 in the marked-up revised manuscript.

2. Please start the abstract by a short introduction of the current problem(s) and the solution, based on the current study, in one or two lines.

Response:

Thank you very much for the kind suggestions. Revised accordingly. The first two lines now introduce the global waste challenge and the proposed EIICB solution.

Changes made in manuscript: Lines 6-15 in the marked-up revised manuscript.

3. What the main significance of paper in comparison is of relates published works?

Response:

Thank you very much for the kind suggestions. Added in both Abstract explaining novelty and superiority over existing designs in the abstract of the revised manuscript.

Changes made in manuscript: Lines 19-23 in the marked-up revised manuscript.

4 &18. Written is very week. In its current state, the level of English throughout the manuscript needs language polishing. Please check the manuscript and refine the language carefully. The language of the manuscript is very pitiful, due to which several times it is hard to understand exactly what the authors want to tell. Hence, the language must be improved by a professional service.

Response:

Thank you very much for the kind suggestions. The authors are very sorry for the English polishing. The entire manuscript has undergone extensive language polishing for clarity, technical precision, and academic tone.

5.The introduction section is very short and poorly described. It doesn't present the reference to the manuscript scope. In the introduction section, authors should make an in-depth literature review concerning the use of new technologies such as green nanotechnology to improve human life, construction and solve environmental and economic problems as well as storage and production of clean energy via simple and economical methods, and sustainability. Introduction has deficiency citation to valuable works published before. The following references are recommended to be cited, to improve the introduction section:

Composites Part B 167 (2019) 643–653; Journal of Environmental Management 350 (2024) 119545; Scientific Reports, (2023) 13:19042, 10.1038/s41598-023-46536-8; Scientific Reports, (2024) 14:10914, 10.1038/s41598-024-61688-x;

Response:

Thank you very much for the kind suggestions. The Introduction is expanded significantly. Relevant references recommended by the Reviewer have been included.

Changes made in manuscript:

Added references are [6], [11], [12], and [37] in lines 56-65 &94 in the marked-up revised manuscript.

6. And the structure of the manuscript might need a major adjusting for a better understanding.

Response:

Thank you very much for the kind suggestions. The manuscript has been reorganized for logical flow.Sections 2-4 are all restructured, with some new subsections added based on the suggestions given by the reviewer.

7. The novelty of this study should be inserted in the text clearly.

Response:

Thank you very much for the kind suggestions. Novelty is now explicitly highlighted in both “Introduction” and “Conclusion” sections (full interlock + insulation + recycled aggregates + mortar-free + full-scale manufacturing).

Changes made in manuscript: The revised parts are in Lines 127-129 and 799-803 in the marked-up revised manuscript.

8 & 13Results and discussion: - To increase the scientific value of the manuscript Authors should consider extension of the all results section with comparison of obtained results with the results described in previous publications. This work should be compared with the other work in Table form.

Response:

Thank you very much for the kind suggestions. The Results and Discussion sections are significantly expanded based on the suggestions. More comparative analysis is added. Table 4 has been improved with clearer formatting and more complete comparisons.

Changes made in manuscript: Line 606 in the marked-up revised manuscript.

9–10. How does your paper contribute to the advancement of knowledge? What are the gap areas and the new contribution in the paper?

Response:

Thank you very much for the kind suggestions. These points are now explicitly described in the Introduction, Discussion, and Conclusion.

Changes made in manuscript: Lines137-142 in the marked-up revised manuscript.

11–12. The authors should prepare all figures with better resolution. The authors should prepare all tables with better quality.

Response:

Thank you very much for the kind suggestions. All figures and tables have been regenerated or enhanced with higher resolution and consistent formatting using NAAS tool. Examples can be seen in Figures 6, 8, and 9.

Changes made in manuscript:

Lines 197, 241, 282, 284, 291, 321-323, 338, 347-349, and 385-387 in the marked-up revised manuscript.

14. Indeed, there are impressive amount of results. However, the conclusions section needs to improve with selected and highlighted main findings.

Response:

Thank you very much for the kind suggestions. The Conclusion has been improved and highlighted key quantitative findings and contributions.

Changes made in manuscript: Lines 807-817 in the marked-up revised manuscript.

15. The discussion is poor and should be improved.

Response:

Thank you very much for the kind suggestions. The Discussion has been expanded with technical interpretation, implications, and literature-supported arguments.

16. Where is the practical application of this manuscript? It must be added.

Response:

Thank you very much for the kind suggestions. Practical applications for housing, mining areas, modular construction, emergency shelters, and reusable building systems have been added. The added subsection 4.6 is for this comment.

Changes made in manuscript: Lines 518-563 in the marked-up revised manuscript.

17. The abstract needs to be revised. Please further organize your views and look to the future.

Response:

Thank you very much for the kind suggestions. A future-direction statement has been added in the Abstract.

Changes made in manuscript: lines 33-36 in the marked-up revised manuscript.

19. The authors are recommended to be cited important references related to using environmentally friendly technologies to solve environmental problems, in the field of health, wellness, treatment, and improving people's lives, environmental remediation, and achieve sustainable development, such as: Results in Engineering 23 (2024) 102399; Results in Engineering 24 (2024) 103281; Results in Engineering 25 (2025) 103758

Response:

Thank you very much for the kind suggestions. The authors have added the recommended references in the Introduction.

Changes made in manuscript: The new numbers of the added references are [34], [35], and [36] in line 94 in the marked-up revised manuscript

REVIEWER #2 COMMENTS

1. The word/property of insulation is very focused in the paper it would be great if this could be added in the title

Response:

Thank you very much for the kind suggestions. The title has been revised to: “Investigations of a new EPS-insulated mortar-free reusable interlocking block for sustainable buildings” based on the suggestion of the reviewer.

2. There is no paper from 2025 which seems to be concerning for a submission in the last quarter of the year

Response:

Thank you very much for the kind suggestions. Recent 2025 references included where relevant.

Changes made in manuscript: The new numbers of the added references are [1], [7], [13], and [36] in lines 50, 57, 65, and 94 in the marked-up revised manuscript.

3. Figure no 1 may be labelled properly with a comprehensive caption

Response:

Thank you very much for the kind suggestions. Fig 1’s caption was expanded, and the labels were corrected.

Changes made in manuscript: Line 198 in the marked-up revised manuscript.

4. Clarify the thermal R-value: State clearly whether the R-value is measured or calculated. If measured, provide full test method, standard apparatus, and raw data. If calculated, provide formulas, thermal conductivity values, assumptions, and uncertainty.

Response:

Thank you very much for the kind suggestions. Updated section explaining that the R-value is calculated in revised Section 3.4, based on EPS thermal conductivity values (the R-value depended on the thickness and density of EPS core used, which was determined directly from the reference) , and assumptions, with supporting reference.

Changes made in manuscript: Lines 393-400 in the marked-up revised manuscript.

5. Provide details/references for the CO₂ reduction claim

Response:

Thank you very much for the kind suggestions. Clarified the basis for the 73.5% CO₂ reduction claim.

Changes made in manuscript: Lines 405-412 in the marked-up revised manuscript.

6. Fully characterize the recycled aggregates and other materials for reproducibility

Response:

Thank you very much for the kind suggestions. Added description: size distribution, crushing method, origin, and cleaning process.

Changes made in manuscript: Lines 167 -170 in the marked-up revised manuscript.

7. The mix design may be written in standard engineering format

Response:

Thank you very much for the kind suggestions. Revised to show ratio, units, and batching method in standard format.

Changes made in manuscript: Lines 176-183 in the marked-up revised manuscript.

8. The authors are encouraged to describe and standardize the compaction method for getting reproducible details (number of tamps, applied energy, or vibration method).

Response:

Thank you very much for the kind suggestions. Compaction procedure clarified.

Changes made in manuscript: Lines 230-233 in the marked-up revised manuscript.

9. How to 2 cm clearance on all sides was maintained?

Response:

Thank you very much for the kind suggestions. Added explanation that spacer blocks and measurement gauges were used.

Changes made in manuscript: Lines 224-229 in the marked-up revised manuscript.

10. How? Cure the blocks for one week and then submerge for 21 days

Response:

Thank you very much for the kind suggestions. Rewritten to explain sequence: 7 days moist curing + 21 days water curing per ASTM recommendations.

Changes made in manuscript: Lines 245-249 in the marked-up revised manuscript.

11. Figure 3a may be appropriately labelled

Response:

Thank you very much for the kind suggestions. Label corrected.

Changes made in manuscript: Line 242 in the marked-up revised manuscript.

12. The authors are encouraged to add references about the quantitative insulating properties of EPS and theirs effect in the current study

Response:

Thank you very much for the kind suggestions. New references added.

Changes made in manuscript: The n

---

## [Decision Letter · Decision Letter 1]

21 Jan 2026

Investigations of a new EPS-insulated mortar-free reusable interlocking block for sustainable buildings

PONE-D-25-54505R1

Dear Dr. Karani,

We’re pleased to inform you that your manuscript has been judged scientifically suitable for publication and will be formally accepted for publication once it meets all outstanding technical requirements.

Kind regards,

Pratheep K. Annamalai

Academic Editor

PLOS One

Additional Editor Comments (optional):

Thanks for addressing the comments. In the experimental section, manual-like bullet point procedure can be rewritten into paragraphs as narrative.

Rest looks fine.

Reviewers' comments:

Reviewer's Responses to Questions

**Comments to the Author**

Reviewer #1: All comments have been addressed

Reviewer #2: All comments have been addressed

2. Is the manuscript technically sound, and do the data support the conclusions?

Reviewer #1: Yes

Reviewer #2: Yes

3. Has the statistical analysis been performed appropriately and rigorously?

Reviewer #1: Yes

Reviewer #2: Yes

4. Have the authors made all data underlying the findings in their manuscript fully available?

Reviewer #1: Yes

Reviewer #2: Yes

5. Is the manuscript presented in an intelligible fashion and written in standard English?

Reviewer #1: Yes

Reviewer #2: Yes

Reviewer #1: The authors make moderate revisions to the manuscript, and give a relevant response to the issues the reviewers concerned. Therefore, it could be considered as potential publication.

Reviewer #2: The authors have addressed the raised shortcoming and have done necessary modifications. The revisions are now satisfactory and i find the manuscript suitable for publications. i have recoomeneded acceptance. Thank you.

**Do you want your identity to be public for this peer review?** For information about this choice, including consent withdrawal, please see our Privacy Policy

Reviewer #1: No

Reviewer #2: No

---

## [Editor Report · Acceptance letter]

17 Nov 2025

PONE-D-25-54505R1

PLOS One

Dear Dr. Karani,

I'm pleased to inform you that your manuscript has been deemed suitable for publication in PLOS One. Congratulations! Your manuscript is now being handed over to our production team.

Kind regards,

on behalf of

Dr. Pratheep K. Annamalai

Academic Editor

PLOS One